# Optical Nonreciprocity Based on the Four-Wave Mixing Effect in Semiconductor Quantum Dots

**DOI:** 10.3390/nano15050380

**Published:** 2025-03-01

**Authors:** Zelin Lin, Han Yang, Fei Xu, Yihong Qi, Yueping Niu, Shangqing Gong

**Affiliations:** 1School of Physics, East China University of Science and Technology, Shanghai 200237, China; zelinlin5@gmail.com (Z.L.); 19821825751@163.com (H.Y.); xfei@ecust.edu.cn (F.X.); niuyp@ecust.edu.cn (Y.N.); sqgong@ecust.edu.cn (S.G.); 2School of Materials Science and Engineering, East China University of Science and Technology, Shanghai 200237, China

**Keywords:** optical nonreciprocity, quantum dot, four-wave mixing, optical isolator

## Abstract

Optical nonreciprocity and nonreciprocal devices such as optical diodes have broad and promising applications in various fields, ranging from optical communication to signal process. Here, we propose a magnet-free nonreciprocal scheme based on the four-wave mixing (FWM) effect in semiconductor quantum dots (SQDs). Via controlling the directions of the coupling fields, the probe field can achieve high transmission in the forward direction within a certain frequency range due to the FWM effect. And the transmission of the probe field in the backward direction undergoes significant reduction, as the FWM effect is absent. The calculation results show a wide nonreciprocal transmission window with isolation greater than 12 dB and insertion loss lower than 0.08 dB. The influences of the Rabi frequencies of the coupling fields, the medium length, and the decay rates on the nonreciprocal propagation of the probe field are also studied, showing the requirements of these parameters for good nonreciprocal performances. Our work may offer an insight for developing optical nonreciprocal devices based on the FWM process and the SQD system.

## 1. Introduction

Optical nonreciprocity is an optical physical process that breaks the Lorentz reciprocity principle and plays a significant role in optical communication and signal processing, etc. [1,2,3]. Currently, realization of optical nonreciprocity is generally based on the traditional magneto-optical Faraday effect [4,5]. However, the magneto-optical Faraday effect is relatively weak in most magneto-optical media, and the incompatibility between magneto-optical materials and silicon-based materials also leads to the bottlenecks in miniaturization and integration [6]. To avoid these limitations, schemes for achieving magnet-free optical nonreciprocity have been proposed in recent years, such as optical nonlinearity [7,8,9,10,11,12], chiral quantum optics [13,14], optomechanical interaction [15,16,17,18], ‘moving’ photonic crystals [19,20,21], and atomic thermal motion [22,23,24,25,26].

Four-wave mixing (FWM) is a typical nonlinear optical effect and has been proposed as a method to achieve optical nonreciprocity. Song et al. explored the realization of optical nonreciprocity using the FWM effect in a thermal atomic system and ultimately gained the desirable nonreciprocal effect with high isolation and low loss [27]. Yang et al. proposed utilizing a double Λ-type thermal atomic system to achieve nonreciprocal amplification [28]. Through the far-detuned FWM process in hot atoms, they experimentally demonstrated an ultra-strong forward gain of 45 dB, while maintaining near-unity transmission in the backward direction. Ge et al. introduced a design approach for optical isolators and circulators, utilizing the enhanced FWM effect in semiconductor double quantum wells [29].

On the other hand, semiconductor quantum dots (SQDs) are nanoscale solid-state semiconductor materials with superior performance. Due to their tunable atom-like properties, strong nonlinear optical coefficients, and flexibility in material composition and fabrication [30], SQD materials are widely used in optical and electronic fields, including high-performance displays, bio-imaging, solar cells, photocatalysis, and sensors. Realizing optical nonreciprocal devices based on SQDs is of great significance for photon integration. However, magnet-free optical nonreciprocity and isolators are still rarely studied in SQDs. In this paper, we propose a scheme to achieve magnet-free optical nonreciprocity based on a direction-dependent phase-matched FWM process in a GaAs/AlGaAs SQD structure. The results show that, by selecting appropriate optical parameters, a nonreciprocal window with the isolation exceeding 10 dB and insertion loss less than 0.08 dB can be obtained. Meanwhile, utilizing the tunable advantage of SQDs, we also discuss the effects of the Rabi frequencies of the coupling and driving fields, penetration depth, and decay rate to achieve further enhancements in nonreciprocal performance.

Although previous work has studied optical nonreciprocity based on the FWM effect in quantum wells and atomic systems, quantum dot systems have more advantages in tunability, integration, and nonlinear response. Unlike quantum wells, where energy levels are less discrete and tunability is limited, SQDs exhibit strong quantum confinement effects, enabling precise control over their optical properties (e.g., emission wavelength and nonlinear response) through size and composition adjustments. This level of tunability is not achievable in quantum wells or atomic systems. Additionally, SQDs are inherently compatible with semiconductor fabrication technologies, making them highly suitable for integration into on-chip photonic devices such as photonic integrated circuits (PICs). This is a significant advantage over atomic systems, which require complex setups (e.g., vacuum chambers and precise laser cooling) and are challenging to integrate into practical applications. Furthermore, the discrete density of states in SQDs leads to stronger nonlinear optical effects, including FWM, compared to quantum wells, allowing for efficient FWM at lower power levels. This makes SQDs more suitable for low-power applications in quantum communication and information processing.

## 2. Theoretical Model and Equations

We consider a GaAs/AlGaAs SQD system, whose energy levels and laser field coupling scheme are depicted in Figure 1. The energy levels |0〉, |1〉, |2〉, and |3〉 are the ground state, two intermediate states, and the excited state [31], respectively. Two strong coupling fields with Rabi frequencies $\Omega_{c}$, $\Omega_{d}$, and detuning $\Delta_{c}$, $\Delta_{d}$, couple the transitions |1〉→|3〉 and |2〉→|3〉, respectively. A weak probe field with frequency $\omega_{p}$, Rabi frequency $\Omega_{p}$, and detuning $\Delta_{p}$ couples the transition |0〉→|1〉. When the probe field propagates in the forward direction, the phase-matched condition would be satisfied. Hence, a FWM field with the frequency $\omega_{m}$ and Rabi frequency $\Omega_{m}$, which couples the transition |0〉→|2〉, will be generated based on the FWM effect (as shown in Figure 1a). Under the electric-dipole and rotating-wave approximations, the effective interaction Hamiltonian of the system in the interaction picture takes the following form [32]:(1)$$H_{I}^{f}=-\left(\begin{array}{cccc} 0 & \Omega_{p}^{f} & \Omega_{m}^{f} & 0\\ \Omega_{p}^{f} & -\Delta_{p} & 0 & \Omega_{c}\\ \Omega_{m}^{f} & 0 & -\Delta_{m} & \Omega_{d}\\ 0 & \Omega_{c} & \Omega_{d} & -(\Delta_{p}+\Delta_{c}) \end{array}\right),$$
where $\Delta_{m}$ is the detuning of the FWM field. The relation $\Delta_{m}=\Delta_{c}+\Delta_{p}-\Delta_{d}$ is satisfied according to the phase-matched condition.

The wavefunction of the system can be defined as $|\Psi \rangle =C_{0}^{f}|0\rangle +C_{1}^{f}|1\rangle +C_{2}^{f}|2\rangle +C_{3}^{f}|3\rangle$, with $C_{j}^{f}\left(j=0,1,2,3\right)$ being the probability amplitude of the energy level |j〉 in the case of the forward probe field. By using the Schrödinger equation $i\frac{\partial }{\partial t}|\Psi \rangle =H_{\mathrm{int}}^{I}|\Psi \rangle$, the motion equations of the probability amplitudes can be obtained as follows,(2)$$\begin{array}{cc} i\frac{\partial }{\partial t}C_{0}^{f} & =-\Omega_{p}^{f}C_{1}^{f}-\Omega_{m}^{f}C_{2}^{f},\\ i\frac{\partial }{\partial t}C_{1}^{f} & =\Gamma_{1}C_{1}^{f}-\Omega_{p}^{f}C_{0}^{f}-\Omega_{c}C_{3}^{f},\\ i\frac{\partial }{\partial t}C_{2}^{f} & =\Gamma_{2}C_{2}^{f}-\Omega_{m}^{f}C_{0}^{f}-\Omega_{d}C_{3}^{f},\\ i\frac{\partial }{\partial t}C_{3}^{f} & =\Gamma_{3}C_{3}^{f}-\Omega_{c}C_{1}^{f}-\Omega_{d}C_{2}^{f}, \end{array}$$
where $\Gamma_{1}=\Delta_{p}-i\gamma_{1}$, $\Gamma_{2}=\Delta_{m}-i\gamma_{2}$, and $\Gamma_{3}=\Delta_{p}+\Delta_{c}-i\gamma_{3}$, and $\gamma_{1}$, $\gamma_{2}$, and $\gamma_{3}$ represent the decay rates corresponding to the energy levels. With the restriction of the weak probe field and FWM field, the majority of electrons reside in the ground state; therefore, we can assume that $|C_{0}^{f}{|}^{2}\approx 1$. Under the steady-state condition, i.e., $i\frac{\partial }{\partial t}C_{j}^{f}=0$, from Equations (1) and (2), the probability amplitude solutions for energy levels |1〉 and |2〉 can be obtained as follows:(3)$$\begin{array}{c} C_{1}^{f}=\frac{D_{1}\Omega_{p}^{f}+D_{2}\Omega_{m}^{f}}{D_{0}},\\ C_{2}^{f}=\frac{D_{3}\Omega_{p}^{f}+D_{4}\Omega_{m}^{f}}{D_{0}}, \end{array}$$
where $D_{0}=\Gamma_{1}\Gamma_{2}\Gamma_{3}-\Gamma_{2}\Omega_{c}^{2}-\Gamma_{1}\Omega_{d}^{2}$, $D_{1}=\Gamma_{2}\Gamma_{3}-\Omega_{d}^{2}$, $D_{2}=D_{3}=\Omega_{c}\Omega_{d}$, and $D_{4}=\Gamma_{1}\Gamma_{3}-\Omega_{c}^{2}$.

It should be noted that the generation of FWM needs to meet the phase-matching conditions. The sensitivity of the phase-matching condition has been extensively discussed in previous work [27,33]. For example, the impact of polarization and a small angle deviation on FWM was also discussed in [27] and [33], respectively. Meanwhile, for the case of the probe field in the backward direction, FWM does not appear due to the phase mismatch (as shown in Figure 1b). Hence, the effective Hamiltonian for the SQD system is given by(4)$$H_{I}^{b}=-\left(\begin{array}{cccc} 0 & \Omega_{p}^{b} & 0 & 0\\ \Omega_{p}^{b} & -\Delta_{p} & 0 & \Omega_{c}\\ 0 & 0 & 0 & \Omega_{d}\\ 0 & \Omega_{c} & \Omega_{d} & -(\Delta_{p}+\Delta_{c}) \end{array}\right).$$

According to Equation (Equation 4), the motion equations of the probability amplitudes can be obtained as follows:(5)$$\begin{array}{cc} i\frac{\partial }{\partial t}C_{0}^{b} & =-\Omega_{p}^{b}C_{1}^{b},\\ i\frac{\partial }{\partial t}C_{1}^{b} & =\Gamma_{1}C_{1}^{b}-\Omega_{p}^{b}C_{0}^{b}-\Omega_{c}C_{3}^{b},\\ i\frac{\partial }{\partial t}C_{2}^{b} & =\Gamma_{2}C_{2}^{b}-\Omega_{d}C_{3}^{b},\\ i\frac{\partial }{\partial t}C_{3}^{b} & =\Gamma_{3}C_{3}^{b}-\Omega_{c}C_{1}^{b}-\Omega_{d}C_{2}^{b}, \end{array}$$
where $C_{j}^{b}\left(j=0,1,2,3\right)$ is the probability amplitude of the energy level |j〉 in this case. From Equation (Equation 5), the steady-state solutions can be written as follows:(6)$$C_{1}^{b}=\frac{D_{1}\Omega_{p}^{b}}{D_{0}}.$$

Using the slowly varying amplitude approximation, the evolution of the forward probe field and the FWM field in the SQD medium can be expressed by Maxwell’s equations as follows [34]:(7)$$\begin{array}{cc} \frac{\partial }{\partial z}\Omega_{p}^{f} & =i\kappa_{1}C_{1}^{f}C_{0}^{f*},\\ \frac{\partial }{\partial z}\Omega_{m}^{f} & =i\kappa_{2}C_{2}^{f}C_{0}^{f*}. \end{array}$$

For the backward direction, the probe field satisfies the following equation:(8)$$\frac{\partial }{\partial z}\Omega_{p}^{b}=i\kappa_{1}C_{1}^{b}C_{0}^{b*},$$
where $\kappa_{1}=\frac{2\pi N\omega_{p}{\left|\mu_{01}\right|}^{2}}{\hbar c}$ and $\kappa_{2}=\frac{2\pi N\omega_{m}{\left|\mu_{02}\right|}^{2}}{\hbar c}$ are the propagation coefficients, with *N* being the electron number density of the SQDs, $\mu_{01(2)}$ the dipole matrix element between levels |0〉 and |1〉 (|2〉), and *c* the speed of light in a vacuum. From Equations (3) and (6)–(8), the forward and backward probe fields after passing through the SQD medium of length *L* are determined by(9)$$\begin{array}{c} \Omega_{p}^{f}\left(L\right)=\frac{1}{2w_{1}}[e^{i\left(w_{1}+w_{2}\right)w_{3}L}-e^{i\left(w_{2}-w_{1}\right)w_{3}L}+e^{i\left(w_{2}-w_{1}\right)w_{4}L}\\ -e^{i\left(w_{1}+w_{2}\right)w_{4}L}+w_{2}e^{i(w_{2}-w_{1})L}+w_{2}e^{i(w_{1}+w_{2})L}]\Omega_{p}^{f}\left(0\right),\\ \Omega_{p}^{b}\left(L\right)=e^{\frac{iD_{1}\kappa_{1}}{D_{0}}L}\Omega_{p}^{b}\left(0\right), \end{array}$$
where $w_{1}=\sqrt{D_{1}^{2}\kappa_{1}^{2}+D_{4}^{2}\kappa_{2}^{2}+4D_{2}D_{3}\kappa_{1}\kappa_{2}-2D_{1}D_{4}\kappa_{1}\kappa_{2}}$, $w_{2}=D_{1}\kappa_{1}+D_{4}\kappa_{2}$, $w_{3}=\frac{D_{1}\kappa_{1}}{2D_{0}}$, and $w_{4}=\frac{D_{4}\kappa_{2}}{2D_{0}}$.

Then, from Equation (Equation 9), the transmission of the probe field in the two reverse directions can be rewritten in the following form [29]:(10)$$\left(\begin{array}{c} \Omega_{p}^{f}\left(L\right)\\ \Omega_{p}^{b}\left(L\right) \end{array}\right)=\left(\begin{array}{cc} \xi_{p}^{f}e^{i\phi_{p}^{f}} & 0\\ 0 & \xi_{p}^{b}e^{i\phi_{p}^{b}} \end{array}\right)\left(\begin{array}{c} \Omega_{p}^{f}\left(0\right)\\ \Omega_{p}^{b}\left(0\right) \end{array}\right),$$
where $\xi_{p}^{f}e^{i\phi_{p}^{f}}=\frac{1}{2w_{1}}[e^{i\left(w_{1}+w_{2}\right)w_{3}L}-e^{i\left(w_{2}-w_{1}\right)w_{3}L}+e^{i\left(w_{2}-w_{1}\right)w_{4}L}-e^{i\left(w_{1}+w_{2}\right)w_{4}L}+w_{2}e^{i(w_{2}-w_{1})L}$

$+\;w_{2}e^{i(w_{1}+w_{2})L}]$, and $\xi_{p}^{b}e^{i\phi_{p}^{b}}=e^{i\frac{D_{1}\kappa_{1}}{D_{0}}L}$, with $\xi_{p}^{f(b)}$ and $\phi_{p}^{f(b)}$ representing the transmission amplitude and phase shift of the forward (backward) probe field, respectively. When optical nonreciprocity is accomplished by the FWM effect, the numerical results will show $\xi_{p}^{f}\neq \xi_{p}^{b}$ or $\phi_{p}^{f}\neq \phi_{p}^{b}$. In this paper, we mainly concentrate on the nonreciprocal transmission amplitude, i.e., $\xi_{p}^{f}\neq \xi_{p}^{b}$.

## 3. Results and Discussion

In this section, we will present the simulation results obtained based on the theory discussed in the previous section. For simplicity, we assume that $\gamma_{1}=\gamma_{2}=\gamma_{3}=\gamma$, $\Omega_{c}=\Omega_{d}=\Omega$, $\Delta_{c}=\Delta_{d}=\Delta$, and $\kappa_{1}=\kappa_{2}=\kappa$ [34]. Figure 2 shows the results of the nonreciprocal transmission of the probe field versus the detuning of the probe field $\Delta_{p}$. Here, the red solid line and the black dashed line represent the forward and backward cases, respectively. Due to the symmetry of the transmission spectrum, we consider only the case where the detuning $\Delta_{p}\geq 0$. It can be clearly observed that there is an obvious nonreciprocal window between 6 meV and 10 meV. Within this window, the probe field is located between two absorption peaks in both cases. However, the forward probe field is partially converted due to the presence of the FWM effect, leading to low transmission. In contrast, the transmission of the backward probe exhibits nearly no loss. The parameters Ω=10meV, γ=0.054meV, Δ=0, κ=10meV/μm, and L=1.65μm are used in the calculation of Figure 2. Similar SQD parameters have been used in previous work [31,34]. By using self-assembling, such parameters can be achieved in InGaAs/GaAs QDs [35].

To characterize the nonreciprocal effect, we define the isolation and insertion loss as $\eta =-10\log_{10}\left(\frac{\xi_{p}^{f}}{\xi_{p}^{b}}\right)$ and $\alpha =-10\log_{10}\left(\xi_{p}^{b}\right)$, respectively. Figure 3 displays the isolation (a) and insertion loss (b) within the nonreciprocal window. The results show that, while keeping the insertion loss below 0.08 dB, an isolation of over 10 dB, reaching up to a maximum of 24 dB, can be achieved. It is demonstrated that a strong nonreciprocal effect can be achieved while avoiding the waste of optical signal resources. For further illustration, we can make a simple comparison with previous non-magnetic optical nonreciprocal schemes. The atomic thermal motion scheme [22] uses contrast to quantify the nonreciprocal effect, which is comparable to our scheme, both being above 0.8. Compared to the 40 dB isolation reported for the silicon photonic diode [8], our method does not show an advantage in terms of isolation. But, our method significantly outperforms both schemes in terms of insertion loss (0.075 dB), with the atomic thermal motion scheme showing a forward transmission rate of only 0.6, and the silicon photonic diode having an insertion loss greater than 1.6 dB. Additionally, due to the small size of quantum dots, our method is more easily integrated into micro-photonic circuits.

In Figure 4, we study the dependence of the transmission of the probe field on the Rabi frequencies Ω under a large probe detuning $\Delta_{p}=16.5\;\mathrm{meV}$. The results demonstrate that both the forward and backward probe fields exhibit a lossless transmission amplitude when the Rabi frequency is small. This is because the large detuning suppresses the absorption of the detection field. As the Rabi frequency increases from 0.1meV to 10meV, the transmission of the forward detection field decreases from 1 to 0.18, while the backward probe field keeps a high transmission. These results verify that, due to the directionality of the FWM effect, the Rabi frequency has a significantly greater influence on the forward probe field compared to the backward case.

In what follows, we discuss variation in the transmission of the probe field with different penetration depths, which is the product of the propagation coefficient κ and the length of the medium *L*. The results are shown in Figure 5. It is found that the transmission of the forward probe field oscillates with the penetration depth in the forward direction, while the backward one decreases slightly with the increase in the penetration depth. The nonreciprocal effect reaches its maximum at κL=17meV. For this penetration depth, the forward transmission is 0.037, and the backward transmission is 0.985. This implies that the nonreciprocal effect can be enhanced by adjusting the medium length or the propagation coefficient in the system.

According to Refs. [36,37], the decay rate γ of the SQDs may increase with the temperature. Under low temperatures, the decay rate can be reduced to 0.01meV, while at room temperature (300 K), it can increase to 10meV. So, we further study the effects of different temperatures on the nonreciprocal transmission in the system via changing the decay rate γ. As shown in Figure 6, when the decay rate γ increases, the transmission amplitude of the forward probe field rises while that of the backward one decreases. With the increase in the decay rates, the transmission of both the forward and backward probe fields is reduced. However, the FWM effect in the forward case is also diminished, meaning that a smaller portion of the forward probe field will be converted. Therefore, the increase in the decay rate leads to an overall increase in the forward probe field transmission and a decrease in the backward one. So, increasing the decay rates with temperature will lead to the suppression of the nonreciprocal effect. However, this limitation on nonreciprocity is mainly due to the increase in the decoherence effect. There are also many schemes that can use different mechanisms to suppress the decoherence effect, which makes it possible to realize optical nonreciprocity in SQD systems at room temperature.

## 4. Conclusions

In summary, we have theoretically proposed a scheme for achieving magnetic-free optical nonreciprocity through the FWM effect in a SQD system. The FWM effect is highly dependent on the propagation direction of the probe field, which serves as the fundamental principle for achieving optical nonreciprocity in this study. The results demonstrated that the proposed scheme is capable of achieving isolation above 10 dB and insertion loss below 0.08 dB in a frequency bandwidth of nearly 5 meV for optical nonreciprocity in the SQDs. We have also discussed the influences of various optical parameters (penetration depth, Rabi frequency, and temperature) on the optical nonreciprocal performance. This work may provide a reference for experiments on optical nonreciprocity and applications of nonreciprocal devices based on semiconductor SQD systems.

## Figures and Tables

**Figure 1 nanomaterials-15-00380-f001:**
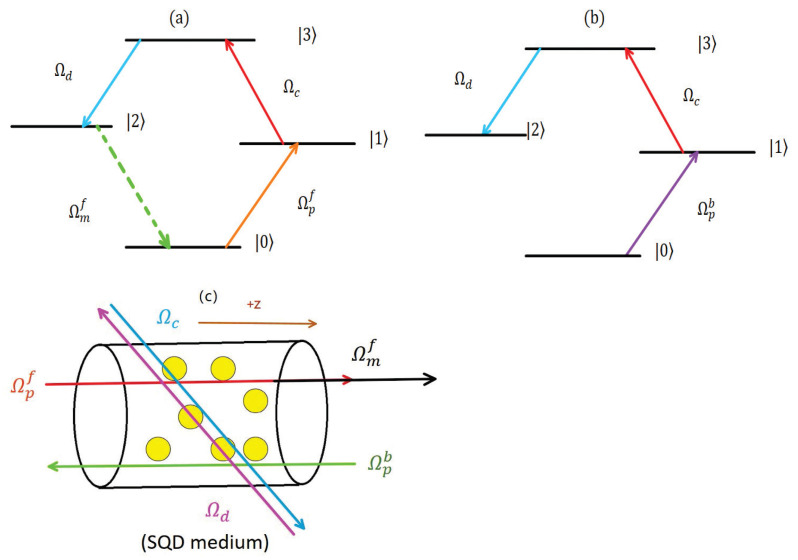
Schematic of the laser coupling schemes and the possible experimental proposal in the GaAs/AlGaAs SQD system. (**a**) Presence of the FWM field when the probe field propagates in the forward direction, (**b**) absence of the FWM field when the probe field propagates in the backward direction, and (**c**) schematic diagram of the interaction between the SQD sample and laser fields.

**Figure 2 nanomaterials-15-00380-f002:**
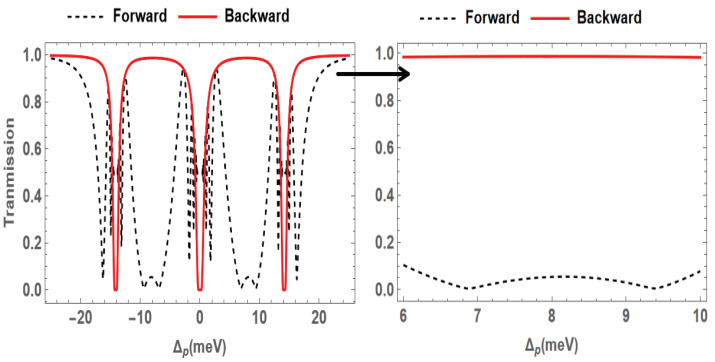
Transmission of the probe field for the forward (red solid) and backward (black dashed) cases. The right figure shows an enlarged view of the nonreciprocal window.

**Figure 3 nanomaterials-15-00380-f003:**
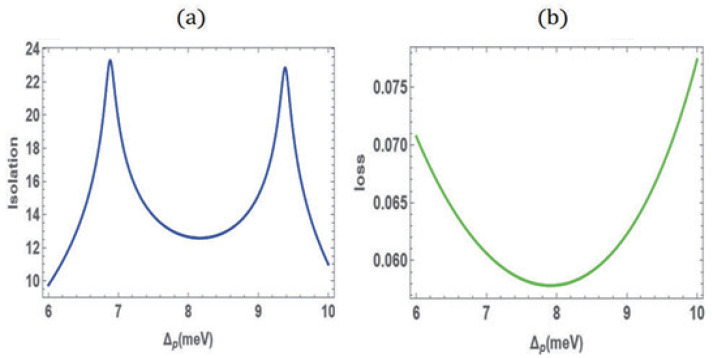
Isolation (**a**) and insertion loss (**b**) within the nonreciprocal window. The parameters used in the calculation are the same as in Figure 2.

**Figure 4 nanomaterials-15-00380-f004:**
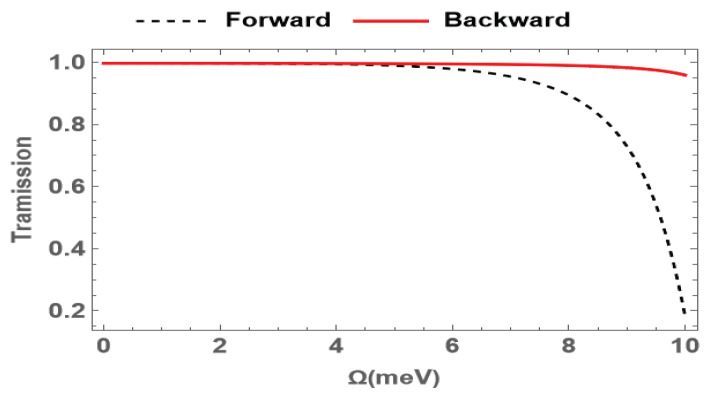
Transmission of the probe field for forward (red solid) and backward (black dashed) cases versus the Rabi frequencies Ω under a large probe detuning $\Delta_{p}=16.5\;\mathrm{meV}$. The other parameters are the same as in Figure 2.

**Figure 5 nanomaterials-15-00380-f005:**
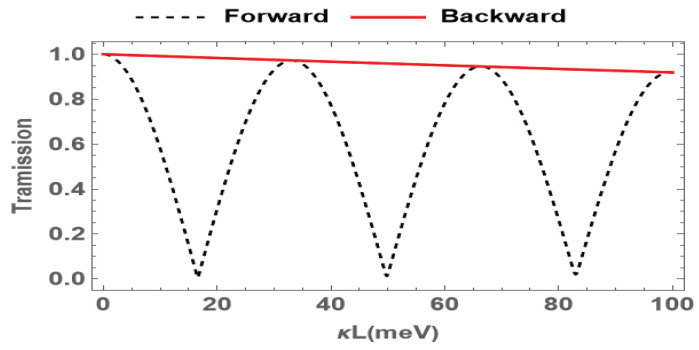
Transmission of the probe field for forward (red solid) and backward (black dashed) cases versus the penetration depths, where the probe detuning is set to be $\Delta_{p}=7$ meV. The other parameters are the same as in Figure 2.

**Figure 6 nanomaterials-15-00380-f006:**
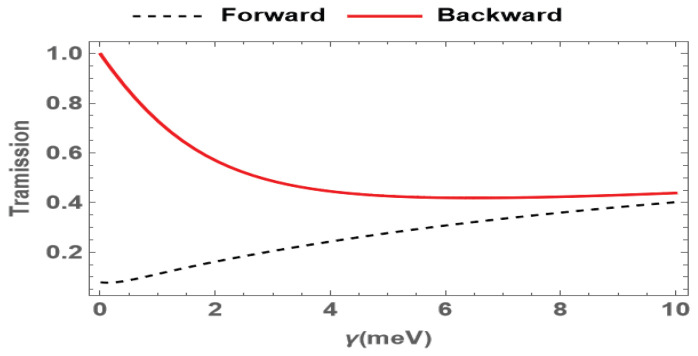
Transmission of the probe field for forward (red solid) and backward (black dashed) cases versus the decay rates, where the probe detuning is set to be $\Delta_{p}=7$ meV. The other parameters are the same as in Figure 2.

## Data Availability

Data are contained within the article.

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
