# Peer review of "Optical Nonreciprocity Based on the Four-Wave Mixing Effect in Semiconductor Quantum Dots"

_nanomaterials, 2025, doi:10.3390/nano15050380_

Round 1
Reviewer 1 Report
Comments and Suggestions for Authors
The manuscript proposes a magnet-free optical nonreciprocal scheme using four-wave mixing (FWM) in semiconductor quantum dots (SQDs), achieving high isolation (>12 dB) and low insertion loss (<0.08 dB). The theoretical framework is detailed and the results are promising, I would like to suggest publication after several critical issues be addressed:
- The novelty of using FWM in SQDs is not sufficiently distinguished from prior work (e.g., FWM in quantum wells [Ref. 29] or atomic systems [Ref. 27]). The authors must explicitly highlight how their SQD-based approach offers unique advantages (e.g., tunability, integration potential) over existing platforms.
- Quantitative comparisons with other magnet-free methods (e.g., nonlinearity-based or optomechanical systems) are missing. For instance, how does the isolation of 24 dB here compare to silicon photonic diodes [Ref. 8] or atomic thermal motion schemes [Ref. 22]?
- Key parameters (e.g., Rabi frequencies Ω = 10 meV, decay rates γ = 0.054 meV) lack experimental validation. Are these values achievable in GaAs/AlGaAs SQDs? References to experimental studies (e.g., Ref. 33 for decay rates) should be expanded to justify feasibility.
- The assumption that γ increases to 10 meV at 300 K raises concerns about room-temperature performance. The authors should discuss whether their scheme remains viable under such conditions or if cryogenic operation is required.
- The phase-matching condition for FWM in the forward direction is pivotal but not rigorously analyzed. How sensitive is the nonreciprocal window to deviations in detuning, alignment, or material defects?
- Terms like "high isolation" and "low loss" should be contextualized. For example, 24 dB isolation is notable but not unprecedented; compared with Ref. 8 (40 dB in silicon).

Author Response
Reference No.: nanomaterials-3466074
Title: Optical nonreciprocity based on the four-wave mixing effect in semiconductor quantum dots
Authors: Zelin Lin, Han Yang, Fei Xu, Yihong Qi, Yueping Niu and Shangqing Gong
Dear referee,
We would like to express our sincere gratitude for your insightful comments on our manuscript. In this revised version, we have carefully addressed all the suggestions and comments, and have made the necessary revisions accordingly. We greatly appreciate the valuable feedback, which has significantly contributed to improving the quality of our work. Below is a summary of our responses and the corresponding changes made in the manuscript.
Response to the comments:
- Comment 1: The novelty of using FWM in SQDs is not sufficiently distinguished from prior work (e.g., FWM in quantum wells [Ref. 29] or atomic systems [Ref. 27]). The authors must explicitly highlight how their SQD-based approach offers unique advantages (e.g., tunability, integration potential) over existing platforms.
Response 1: Thank you for your constructive suggestions on further clarifying the novelty of our work. We have illustrated the unique advantages of using FWM in SQDs compared to previous work in quantum wells or atomic systems in the manuscript. The key distinction lies in the exceptional tunability, integration, and enhanced nonlinear response of SQDs. Unlike quantum wells, where energy levels are less discrete and tunability is limited, SQDs exhibit strong quantum confinement effects, enabling precise control over their optical properties (e.g., emission wavelength and nonlinear response) through size and composition adjustments. This level of tunability is not achievable in quantum wells or atomic systems. Additionally, SQDs are inherently compatible with semiconductor fabrication technologies, making them highly suitable for integration into on-chip photonic devices such as photonic integrated circuits (PICs). This is a significant advantage over atomic systems, which require complex setups (e.g., vacuum chambers and precise laser cooling) and are challenging to integrate into practical applications. Furthermore, the discrete density of states in SQDs leads to stronger nonlinear optical effects, including FWM, compared to quantum wells, allowing for efficient FWM at lower power levels. This makes SQDs more suitable for low-power applications in quantum communication and information processing.
- Comment 2: Quantitative comparisons with other magnet-free methods (e.g., nonlinearity-based or optomechanical systems) are missing. For instance, how does the isolation of 24 dB here compare to silicon photonic diodes [Ref. 8] or atomic thermal motion schemes [Ref. 22]?
Response 2: Thank you for your suggestion on quantitative comparisons with other magnet-free methods, which significantly improves our manuscript. We have provided a detailed comparison in the manuscript. The atomic thermal motion scheme [Ref. 22] uses contrast to quantify the nonreciprocal effect, which is comparable to our scheme, both being above 0.8. Compared to the 40 dB isolation reported for the silicon photonic diode [Ref. 8], our method does not show an advantage in terms of isolation. But, our method significantly outperforms both schemes in terms of insertion loss (0.075 dB), with the atomic thermal motion scheme showing a forward transmission rate of only 0.6, and the silicon photonic diode having an insertion loss greater than 1.6 dB. Additionally, due to the small size of quantum dots, our method is more easily integrated into micro-photonic circuits.
- Comment 3: Key parameters (e.g., Rabi frequencies Ω = 10 meV, decay rates γ = 0.054 meV) lack experimental validation. Are these values achievable in GaAs/AlGaAs SQDs? References to experimental studies (e.g., Ref. 33 for decay rates) should be expanded to justify feasibility.
Response 3: Thank you for reminding us about parameters in the QD system. We use the SQD parameters that has been used as in previous papers [Optik, 2019, 180: 295-301; Journal of the Optical Society of America B, 2012, 29(3): 420-428]. Such parameters of QD can be achieved by using the self-assembled InGaAs/GaAs QD [Phys. Rev. B, 2010, 81(24): 245324]. We have thoroughly enhanced the citation of the sources from which the quantum dot data were obtained.
- Comment 4: The assumption that γ increases to 10 meV at 300 K raises concerns about room-temperature performance. The authors should discuss whether their scheme remains viable under such conditions or if cryogenic operation is required.
Response 4: We are grateful for your insightful comment. Indeed, as the temperature approaches room temperature, the nonreciprocal effect of the entire quantum dot system significantly deteriorates, highlighting the system's sensitivity to temperature. However, this limitation on nonreciprocity is mainly due to the increase of decoherence effect. There are also many schemes that can use different mechanisms to suppress the decoherence effect, which makes it possible to realize optical nonreciprocity in SQD systems at room temperature. We have expanded our discussion in the manuscript to provide a more comprehensive explanation of this aspect.
- Comment 5: The phase-matching condition for FWM in the forward direction is pivotal but not rigorously analyzed. How sensitive is the nonreciprocal window to deviations in detuning, alignment, or material defects?
Response 5: Thank you for your suggestion on discussing the phase-matching condition. Sensitivity of the phase matching condition has been extensively discussed in previous studies. For example, the sensitivity of four-wave mixing to the polarization of forward and backward probe fields was experimentally investigated by rotating a half-wave plate [Appl. Phys. Lett., 2021, 119(2):024101]. Additionally, impact of a small angle between two light fields on the FWM was also discussed in [Adv. Opt. Mater., 2024, 12(35): 2401741]. We have added some illustration in the revised manuscript.
- Comment 6: Terms like "high isolation" and "low loss" should be contextualized. For example, 24 dB isolation is notable but not unprecedented; compared with Ref. 8 (40 dB in silicon).
Response 6: Thank you for pointing out the inappropriate use of terms like "high isolation" and "low loss". As has been mentioned in Response 2, we have analyzed the results and revised relevant terms in the revised manuscript.
In summary, we have taken into account all the suggestions and comments by the referee and have made amendments in the manuscript. Corresponding revisions have been highlighted in red in the manuscript. We hope that the present version of the manuscript can be accepted in EPJD.
Yours sincerely,
Zelin lin and Yihong Qi, on behalf of all the authors
qiyihong@ecust.edu.cn
Reviewer 2 Report
Comments and Suggestions for Authors
The paper by Zelin Lin et al. is devoted to theoretical study of nonreciprocal light propagation due to four-wave mixing (FWM) in semiconductor quantum dot (QD) media. Nonlinearity is one of the most popular approaches to nonreciprocity, so that employing parametric effects such as FWM to realize nonreciprocal transmission is a natural step. The authors utilize the four-level model to describe FWM and discuss how transmission of the probe field depends on the propagation direction at different parameters. The idea is not new and the results are questionable for me, so that I cannot recommend the maniscript in its current form. Further, I give the specific comments to be considered by the authors.
1) As I have mentioned, the idea to use FWM for nonreciprocity is now new. Therefore, it is of primary importance to clearly state what are the distinctive features of QDs as a material for FWM. Does the FWM in QDs differ from the FWM in atomic media? Moreover, I am not sure that the four-level scheme shown in Fig. 1 can be realized in QDs. In particular, the two middle levels (|1> and |2>) have close energies that is quite unusual for the QDs where the gaps between levels are usually of the same order of magnitude.
2) The sourse of asymmetry leading to nonreciprocity should be clearly explained in the paper. Figure 1 shows only the energy conservation, but also the momentum conservation is needed. The authors mention the phase-matching condition, but do not show how it applies in the case discussed.
3) I have great doubts in the correctness of Eqs. (2) and (5) for the description of the system. In particular, there is no relaxation to the ground state (|0>). The ground state is repopulated exclusively due to the induced emission from the upper levels. The relaxation gamma terms result in the particles leaving the system, so that the full probability (the sum of |C|^2 terms) is not equal to 1 and decreases with time. Therefore, I doubt that the steady-state regime is possible in this system at all.
4) Several references are shown just with the "?" marks.
Author Response
Reference No.: nanomaterials-3466074
Title: Optical nonreciprocity based on the four-wave mixing effect in semiconductor quantum dots
Authors: Zelin Lin, Han Yang, Fei Xu, Yihong Qi, Yueping Niu and Shangqing Gong
Dear referee,
We would like to express our sincere gratitude for your insightful comments on our manuscript. In this revised version, we have carefully addressed all the suggestions and comments, and have made the necessary revisions accordingly. We greatly appreciate the valuable feedback, which has significantly contributed to improving the quality of our work. Below is a summary of our responses and the corresponding changes made in the manuscript.
Response to the referee’s comments:
- Comment 1: As I have mentioned, the idea to use FWM for nonreciprocity is not new. Therefore, it is of primary importance to clearly state what are the distinctive features of QDs as a material for FWM. Does the FWM in QDs differ from the FWM in atomic media? Moreover, I am not sure that the four-level scheme shown in Fig. 1 can be realized in QDs. In particular, the two middle levels (|1> and |2>) have close energies that is quite unusual for the QDs where the gaps between levels are usually of the same order of magnitude.
Response 1: Just as you pointed out that, FWM for nonreciprocity have been studied in other materials such as atomic systems and quantum wells. But the SQDs system has its own unique advantages. We have illustrated the unique advantages of using FWM in SQDs compared to previous work in quantum wells or atomic systems in the revised manuscript. The key distinction lies in the exceptional tunability, integration, and enhanced nonlinear response of SQDs. Unlike quantum wells, where energy levels are less discrete and tunability is limited, SQDs exhibit strong quantum confinement effects, enabling precise control over their optical properties (e.g., emission wavelength and nonlinear response) through size and composition adjustments. This level of tunability is not achievable in quantum wells or atomic systems. Additionally, SQDs are inherently compatible with semiconductor fabrication technologies, making them highly suitable for integration into on-chip photonic devices such as photonic integrated circuits (PICs). This is a significant advantage over atomic systems, which require complex setups (e.g., vacuum chambers and precise laser cooling) and are challenging to integrate into practical applications. Furthermore, the discrete density of states in SQDs leads to stronger nonlinear optical effects, including FWM, compared to quantum wells, allowing for efficient FWM at lower power levels. This makes SQDs more suitable for low-power applications in quantum communication and information processing.
Thank you for reminding us about parameters in the QD system. We use the SQD parameters that has been used as in previous papers [Optik, 2019, 180: 295-301; Journal of the Optical Society of America B, 2012, 29(3): 420-428]. Such parameters of QD can be achieved by using the self-assembled InGaAs/GaAs QD [Phys. Rev. B, 2010, 81(24): 245324]. As for question of the close energy separation between the two middle levels (|1> and |2>), it does not affect the generation of FWM effect. In fact, even in a three-level system with a middle level, degenerated FWM effect can also be achieved. We have added a diagram and illustration in the revised manuscript.
- Comment 2: The source of asymmetry leading to nonreciprocity should be clearly explained in the paper. Figure 1 shows only the energy conservation, but also the momentum conservation is needed. The authors mention the phase-matching condition, but do not show how it applies in the case discussed.
Response 2: We appreciate your observation regarding our oversight in addressing the phase-matching condition. We have incorporated the influence of momentum into both the forward and backward Hamiltonians, as exemplified by the interaction term of the probe field (Ωpeikr). Furthermore, we have elaborated on how the differing momentum conservation conditions in the forward and backward directions contribute to the nonreciprocal effects observed. This addition aims to clarify the role of momentum conservation in establishing the asymmetry necessary for nonreciprocity and to provide a more thorough explanation of the phase-matching condition's application within our discussed scenario. We added the corresponding discussion in the revised manuscript.
- Comment 3: I have great doubts in the correctness of Eqs. (2) and (5) for the description of the system. In particular, there is no relaxation to the ground state (|0>). The ground state is repopulated exclusively due to the induced emission from the upper levels. The relaxation gamma terms result in the particles leaving the system, so that the full probability (the sum of |C|^2 terms) is not equal to 1 and decreases with time. Therefore, I doubt that the steady-state regime is possible in this system at all.
Response 3: The density matrix equations and the probability amplitude equations are two commonly methods to study the interaction between quantum medium and laser fields. In this work, we use the probability amplitude equation to solve the problem which is also derived form from the Schrödinger equations. There are some differences in the description of decays between the two methods. But in general, they can get the correct results within the interaction time, which has been confirmed in many previous studies. In the SQD system, the total decay rate for a specific subband can be decomposed into two parts:γj = γjl+γjd. Among them, γjl represents for the lifetime broadening caused by processes such as longitudinal optical (LO) phonon emission, while γjd represents for decoherence. We are discussing under the condition where the control field and the driving field are significantly larger than the probe fieldΩp. Under the steady-state conditions, the majority of electrons reside in the ground state, and the influence of decay on the ground state is relatively minor. Therefore, it is reasonable to approximate the probability amplitude of the ground state as 1, ie. |C0|2≈1.
- Comment 4: Several references are shown just with the "?" marks.
Response 4: Thank you for your careful review and reminding us about the presence of question marks in our references. We have corrected these mistakes in the introduction accordingly.
In summary, we have taken into account all the suggestions and comments by the referee and have made amendments in the manuscript. Corresponding revisions have been highlighted in red in the manuscript. We hope that the present version of the manuscript can be accepted in Nanomaterials.
Yours sincerely,
Zelin lin and Yihong Qi, on behalf of all the authors
qiyihong@ecust.edu.cn

Reviewer 3 Report
Comments and Suggestions for Authors Summary:This paper introduces a magnet-free optical nonreciprocal scheme utilizing the four-wave mixing (FWM) effect in semiconductor quantum dots (SQDs). The approach involves controlling the directions of coupling fields to achieve nonreciprocal transmission of a probe field. The forward transmission exhibits high efficiency due to the FWM effect, while the backward transmission is significantly suppressed. The study demonstrates a wide nonreciprocal transmission window with an isolation greater than 12 dB and an insertion loss of less than 0.08 dB. The authors also explore the influence of parameters such as the Rabi frequencies of coupling fields, medium length, and decay rates on the nonreciprocal performance.
It is unclear why the authors have included so many question marks in the introduction. Is this a misprint, or are they perhaps attempting to cite manuscripts? Additionally, there is no clear reference to how Equation 1 was derived. While the work is purely simulation-based, it would strengthen the manuscript if the authors could include a schematic or setup to demonstrate the experimental feasibility of the proposed scheme. Furthermore, the authors should discuss prior work on nonreciprocity using four-wave mixing in more detail, as several relevant studies appear to be missing. The introduction should also clearly distinguish this work from previous research in the field.
The authors should elaborate on the physical mechanism underlying the four-wave mixing effect in SQDs to make the concept accessible to a broader audience. it would be helpful to provide a more detailed analysis or optimization range for practical implementation.
Comparison with Existing Technologies: Including a comparison with other nonreciprocal schemes (e.g., magnet-based systems or other quantum-dot-based approaches) would contextualize the significance of this work.
The potential limitations or challenges of implementing this scheme in real-world optical systems should be discussed. For instance, how sensitive is the system to parameter variations?
The authors could expand on specific application areas, such as optical communication systems or integrated photonics, where this approach would have the most impact.
grammatical corrections are needed (e.g., "proposes" → "propose," "promissing" → "promising"), in the abstract ....check for the whole manuscript .....
need improvement
Author Response
Reference No.: nanomaterials-3466074
Title: Optical nonreciprocity based on the four-wave mixing effect in semiconductor quantum dots
Authors: Zelin Lin, Han Yang, Fei Xu, Yihong Qi, Yueping Niu and Shangqing Gong
Dear referee,
We would like to express our sincere gratitude for your insightful comments on our manuscript. In this revised version, we have carefully addressed all the suggestions and comments, and have made the necessary revisions accordingly. We greatly appreciate the valuable feedback, which has significantly contributed to improving the quality of our work. Below is a summary of our responses and the corresponding changes made in the manuscript.
Response to the referee’s comments:
- Comment 1: It is unclear why the authors have included so many question marks in the introduction. Is this a misprint, or are they perhaps attempting to cite manuscripts? Additionally, there is no clear reference to how Equation 1 was derived. While the work is purely simulation-based, it would strengthen the manuscript if the authors could include a schematic or setup to demonstrate the experimental feasibility of the proposed scheme. Furthermore, the authors should discuss prior work on nonreciprocity using four-wave mixing in more detail, as several relevant studies appear to be missing. The introduction should also clearly distinguish this work from previous research in the fiel
Response 1: Thank you for reminding us the mistakes about question marks in the introduction. We have corrected these mistakes in the introduction accordingly. Derivation of the Hamiltonian of equation 1 can refer to the relevant books and materials of quantum optics, and we also added a classical reference book in this manuscript. To provide a reference for the experiment, we gave a diagram of the possible experimental scheme in Figure 1 and added corresponding discussion in the text. We have also illustrated the advantages of using FWM to achieve optical nonreciprocity in SQDs compared to previous work in the manuscript.
- Comment 2: The authors should elaborate on the physical mechanism underlying the four-wave mixing effect in SQDs to make the concept accessible to a broader audience. It would be helpful to provide a more detailed analysis or optimization range for practical implementation.
Response 2: We are deeply appreciative of your valuable feedback. In semiconductor quantum dots, the four-wave mixing effect is realized through the nonlinear interaction between the optical fields and the electronic states of the quantum dots, which can generate new frequency components under the influence of pump and probe fields. This effect is contingent upon the energy level structure of the quantum dots, the intensity of the optical fields, and the phase-matching conditions. We have enriched our manuscript with a more thorough and accessible explanation of the FWM effect, aiming to elucidate the underlying physical mechanisms.
- Comment 3: Comparison with Existing Technologies: Including a comparison with other nonreciprocal schemes (e.g., magnet-based systems or other quantum-dot-based approaches) would contextualize the significance of this work.
Response 3: Thank you for your constructive suggestions on further clarifying the novelty of our work. We have illustrated the unique advantages of using FWM in SQDs compared to previous work in the manuscript. The key distinction lies in the exceptional tunability, integration, and enhanced nonlinear response of SQDs.
- Comment 4: The potential limitations or challenges of implementing this scheme in real-world optical systems should be discussed. For instance, how sensitive is the system to parameter variations?
Response 4: We are grateful for your observation regarding our oversight in discussing the limitations of our work. We have given the corresponding discussion in the revised manuscript.
- Comment 5: The authors could expand on specific application areas, such as optical communication systems or integrated photonics, where this approach would have the most impact.
Response 5: Thank you for your suggestion. We have expanded on specific application areas where our work could have a significant impact. For instance, in optical communication systems, the nonreciprocal behavior facilitated by four-wave mixing in quantum dots (QDs) could be utilized to develop optical isolators and circulators without the need for magnetic materials, which would offer a more compact and efficient alternative. This could be particularly valuable in integrated photonics, where reducing size and material complexity is crucial for achieving scalable systems. Moreover, the ability to achieve nonreciprocity in a semiconductor-based platform could enable new functionalities in quantum networks and on-chip photonic circuits, allowing for enhanced control over light propagation and interaction. We hope that these additions provide a clearer understanding of the broader relevance and impact of our work.
- Comment 6: grammatical corrections are needed (e.g., "proposes" → "propose," "promissing" → "promising"), in the abstract ....check for the whole manuscript .....
Response 6: Thanks for your careful review and pointing out the grammar mistake on our manuscript. We have carefully checked and revised our manuscript.
In summary, we have taken into account all the suggestions and comments by the referee and have made amendments in the manuscript. Corresponding revisions have been highlighted in red in the manuscript. We hope that the present version of the manuscript can be accepted in Nanomaterials.
Yours sincerely,
Zelin lin and Yihong Qi, on behalf of all the authors
qiyihong@ecust.edu.cn
Round 2
Reviewer 2 Report
Comments and Suggestions for Authors
I am satisfied with the authors' response, so that the paper can now be recommended for publication.